# At-scale Model Output Statistics in mountain environments (AtsMOS v1.0)

Maximillian Van Wyk de Vries[1,2,3], Tom Matthews[4], L. Baker Perry[5], Nirakar Thapa[6], and Rob Wilby[7]

[1]Department of Geography, University of Cambridge, Cambridge CB2 3EL, UK.
[2]Department of Earth Sciences, University of Cambridge, Cambridge CB3 0EZ, UK.
[3]School of Geography and the Environment, University of Oxford, Oxford OX1 3QY, UK.
[4]Department of Geography, Kings College London, London WC2B 4BG, UK.
[5]Department of Geography and Planning, Appalachian State University, Boone, North Carolina, USA.
[6]Department of Hydrology and Meteorology, Kathmandu, Nepal.
[7]Department of Geography and Environment, Loughborough University, Loughborough, UK

**Correspondence:** M. Van Wyk de Vries (msv27@cam.ac.uk)

**Abstract.** This paper introduces the AtsMOS workflow, designed to enhance mountain meteorology predictions through the downscaling of coarse numerical weather predictions using local observational data. AtsMOS provides a modular, open-source toolkit for local and large-scale forecasting of various meteorological variables through modified Model Output Statistics – and may be applied to data from a single station or an entire network. We demonstrate its effectiveness through an example application at the summit of Mt. Everest, where it improves the prediction of both meteorological variables (e.g., wind speed, temperature) and derivative variables (e.g., facial frostbite time) critical for mountaineering safety. As a bridge between numerical weather prediction models and ground observations, AtsMOS contributes to hazard mitigation, water resource management, and other weather-dependant issues in mountainous regions and beyond.

## 1 Introduction

Accurate mountain weather forecasts facilitate improved hazard mitigation for the 300 million mountain inhabitants worldwide and contribute to effective sustainable resource management (e.g., Miner et al., 2020; Corbari et al., 2022). Furthermore, they are relevant to the 1.6 billion who live downstream of mountains, and depend on their supply of freshwater or are susceptible to their hazards (Immerzeel et al., 2020). However, producing skilful forecasts in such environments is challenging. Major topographic variations cause large spatial variability in the weather, meaning that reality can diverge substantially from Numerical Weather Prediction (NWP) grid-point forecasts within typical $10^2$-$10^3$ km$^2$ grid-cell areas (Zhang et al., 2022). Whilst consistent biases can be adjusted for (e.g., mismatches in elevation between forecast grid points and land surface locations of interest with knowledge of the lapse rate; Minder et al., 2010), the impact of unresolved processes – for instance, local valley or glacier winds driven by surface heat fluxes (Khadka et al., 2022) – is challenging to correct for a priori.

Although advances in NWP, such as finer grid resolutions and refinement of physical parameterisation schemes, may enhance forecast performance in mountainous terrain, progress can be costly and slow (Bauer et al., 2015). A cheaper, faster, and more flexible option to improve forecasts for target locations is to statistically post-process NWP output through calibration to

observations. Model Output Statistics (MOS), which applies multiple linear regression to adjust forecast fields, has historically been the most popular method in this regard (Glahn and Lowry, 1972; Glahn, 2014). In particular, MOS can be used to create forecasts of variables (predictands) not available in NWP model output (Rasp et al., 2020). Recent advancements in

computational power have enabled machine learning to improve the performance of weather forecasts (Lam et al., 2023), including through post-processing (e.g., Lagerquist et al., 2017; Herman and Schumacher, 2018; Han et al., 2021; Grönquist et al., 2021).

However, practical barriers may limit the uptake of such developments at scale. For example, without reference workflows to facilitate the non-trivial task of accessing and pre-processing large NWP datasets forecasts improved by machine learning

are unlikely to reach the diverse range of potential end users in mountainous environments (Table 1). The benefits of highly accurate local weather predictions for use in other (e.g., hydrological) modelling chains may not be achieved if such forecasts are not made available in an interoperable format that follows well-known conventions, such as the 'CF' – Climate and Forecast convention (Eaton et al., 2023).

**Table 1.** Examples of weather variables (predictands) and sectors in which highly accurate, site-specific forecasts may be desirable. [1]The term mountaineering is used to represent a wider set of similar activities – e.g., hiking, skiing and climbing.

| Predictand | Example sector(s) |
|---|---|
| Precipitation amount and phase | Hazard forecasting (flood, avalanche); resource planning |
| Maximum wind gust | Aviation; [1]mountaineering; hazard (avalanche) forecasting |
| Ground temperature | (Road) transport; mountaineering |
| Wind chill temperature | Mountaineering |
| Cloud base and cloud top | Aviation; mountaineering |
| Probability of rime ice accretion | Communications |
| Facial frostbite time | Mountaineering |

Hence, our paper aims to introduce a user-friendly, lightweight version of MOS to fill this gap. We describe modular Python

code that calibrates and applies MOS, including state-of-the-art machine learning algorithms, to produce corrected forecasts in an interoperable format that can feed into other automated workflows to enable at-scale MOS. We anticipate that these features of AtsMOS will, combined with efforts to improve the availability of high-altitude weather observations worldwide (GEO Mountains 2022), offer a step change in the ability to forecast critical mountain weather variables.

In Section 2 we describe the main features of AtsMOS, before illustrating its use in forecasting the weather on the summit

of Mt. Everest, where highly accurate predictions can be the difference between life and death (Section 3). In Section 4 we discuss opportunities and challenges in using AtsMOS more broadly.

## 2 The AtsMOS workflow

### 2.1 Workflow overview

AtsMOS is designed to be a computationally light and flexible template. It has (i) a flexible loading and preprocessing module, which draws in external data, deals with erroneous or missing data and prepares it for further analysis. Our code here is intended as a template such that users may set up their own data loading and pre-processing as the need arises. We do not, therefore, describe each operation in detail but instead refer readers to the documented AtsMOS jupyter notebook associated with this paper. (ii) A core processing module, comprising a modular suite of statistical and machine learning techniques to calibrate and perform data corrections, with XGBoost being the default and most advanced option. (iii) A post-processing module to calculate derivative variables and export the data in the self-describing and interoperable MDF format (GEO Mountains, 2022; Figure 9).

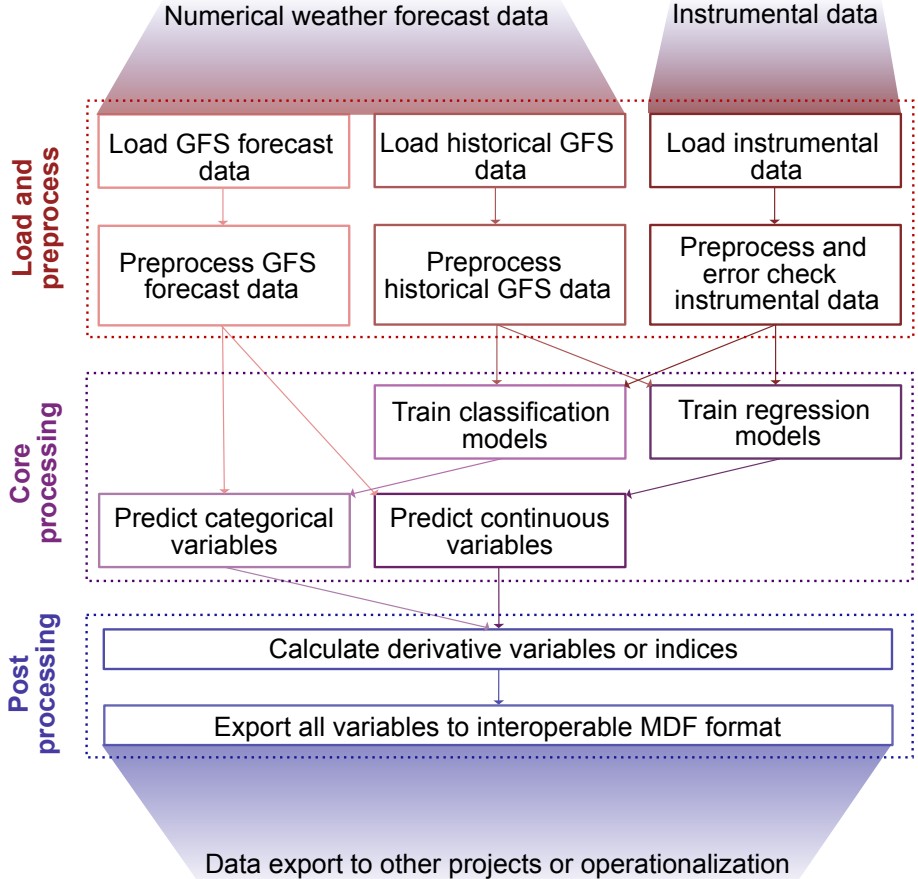

**Figure 1.** Overview diagram of the AtsMOS workflow. The loading of historical GFS data and loading and preprocessing of instrumental data are flexible components subject to user modification, while the others are fixed in this workflow.

## 2.2 Data access and preprocessing

AtsMOS is currently designed to be used with data from the Global Forecasting System (GFS) data from the US National Centre for Atmospheric Research, which is freely available on a global scale and real-time basis (https://rda.ucar.edu/datasets/ds084.1/).
Alternative global or regional numerical prediction models, such as those produced by the European Centre for Medium-Range Weather Forecasts (ECMWF) or national meteorological agencies would also be suitable, particularly if real-time data can be accessed through an Application Programming Interface (API).

GFS forecasts are computed every 6 hours, with a lead time from 0 to 384 hours (16 days). Pressure-level data are generally preferred for mountain forecasting applications because the real-world surface in such regions is likely to be very different (e.g.,
in elevation and surface type) from the model surface (Mass et al., 2008), and hence we anticipate greater general predictability using data from the free atmosphere. We evaluated different methods for accessing GFS data and found that the web subsetting form is generally the most convenient for accessing historical archive data (rda.ucar.edu/datasets/ds084.1/dataaccess/), while the online THREDDS server is best for downloading real-time data (https://tds.scigw.unidata.ucar.edu/thredds/catalog/idd/forecastModels.h As such we include a preset module in AtsMOS for both automatically downloading and pre-processing real-time data, but
only apply the pre-processing for the historical archive data. Historical archive data need only be downloaded once for pre-trained MOS models to be created (see below), which may then be run on any real-time data, ensuring the model remains computationally efficient.

We include example scripts in the AtsMOS jupyter notebook used to preprocess instrumental data (Figure 2). For instance, it is often necessary to synchronise measurement and NWP measurement measurement timings - in the Everest example
presented below instrumental data has a higher measurement frequency than the GFS data (6-hours). We also include scripts for error-checking and filtering of unreliable data, although we note that these are heavily dependent on the type and location of the sensor. We encourage users to carefully consider what if any, processing steps are necessary to field data treated as 'ground truth', as any errors or biases remaining will be learnt by the model. We discuss this in further detail in our limitations section.

## 2.3 Core machine learning

For the core processing, AtsMOS applies Model Output Statistics (MOS) to the GFS data, with a range of possible correction algorithms for the user to select from depending on predictand type (e.g., binary or continuous) and the weighting of interpretability versus performance (Table 2). In our case study below (Section 3), we compare the results from applying simple linear regression and XGBoost (Chen and Guestrin, 2016). Linear regression works well when the relationship between the predictor and target variable is approximately linear. Its coefficients provide clear insights into the impact of each feature,
making it valuable for tasks where interpretability is crucial. However, linear regression cannot resolve nonlinear relationships in the data (without transformations to the input variables) and is sensitive to data quality and outliers limiting its predictive performance in many real-world cases. We implement a standard ordinary least-squares-based linear regression algorithm in AtsMOS, which does not require any parameter choices.

XGBoost is at the other end of the complexity spectrum, combining decision trees with gradient boosting to improve computational efficiency and predictive performance, particularly in high-dimensional, nonlinear data scenarios. It has been shown to outperform most methods in terms of predictive accuracy (Chen and Guestrin, 2016) and is robust to overfitting, but its complexity can make it less suitable in cases where model transparency is essential and understanding the reasons behind incorrect predictions is key. A range of different parameters in XGBoost can be modified from their default values, including the type of objective function used (here, squared error), the learning rate (here, 0.1), the number of estimators (here, 250), the maximum tree depth (here, 4) and more (Chen and Guestrin, 2016). There is no parameter set that is optimal for all datasets. We tune the default parameters for AtsMOS based on the Mt Everest case study described in section 3, which we expect to be broadly applicable (if not optimal) for a wide range of cases. Users may easily run custom hyperparameter tuning for other custom datasets, but we do not include this in the default workflow due to its high computational cost.

AtsMOS is designed to be modular, such that users can easily define new core ML processing algorithms where either methodological advances or specifics of their dataset demand a different approach. We implement linear regression, Random Forest, and XGBoost algorithms, and note that most alternative ML techniques implemented in the scikitlearn python package can also be used by changing a single line of code. An overview of the advantages and limitations of the different models implemented is provided in Table 2. Beyond the initial model evaluation (discussed in section 3.3), we do not split the instrumental data into testing and training data. Instead, we train the ML model of choice using the full instrumental record to maximise both the volume and diversity of training data. We train a separate ML model for each forecast lead time as data error is expected to vary with lead time. All pre-trained ML models are saved once training is completed, and can be directly loaded from file for future AtsMOS runs. In cases where no new instrumental data is available, this enables highly-efficient runs in which model training can be bypassed entirely (Figure 2). Where novel instrumental data is regularly available, we recommend periodic re-training of ML models to maximise expected forecast skill.

## 2.4  Post processing and validation metrics

Once the appropriate ML model has been trained using the historical data, AtsMOS can process the real-time GFS forecast to produce corrected forecasts. The calibrated forecast may be a continuous variable (e.g. wind speed), a probability (e.g. probability of winds above a given threshold), or a binary categorized field (e.g. winds above or below a given threshold) depending on the processing choices made and project requirements. We also highlight that the flexible approach of AtsMOS enables prediction of any variable for which observations exist, and which are sensitive to the atmospheric state. We showcase this in Section 3, making predictions for facial frostbite time – an important variable for mountaineering, which is not available as a direct output from any NWP model.

We use a range of possible metrics to evaluate model performance, including three primary metrics: Kling-Gupta Efficiency (KGE; Gupta et al., 2009), Mean Absolute Error (MAE), and Root Mean Square Error (RMSE). These metrics provide a comprehensive assessment of the model's accuracy, precision, and overall performance. KGE offers a balanced evaluation by combining correlation, bias, and variability, making it particularly suitable for hydrological and meteorological applications. MAE measures the average magnitude of errors, providing a straightforward interpretation of forecast accuracy. RMSE, on

**Table 2.** Different machine learning algorithms currently implemented in AtsMOS, along with strengths and weaknesses. AtsMOS is modular, and users can easily define new core processing algorithms.

| Algorithm | Description | Strengths | Weaknesses |
|---|---|---|---|
| **Linear regression** | Regression using ordinary least square fit | Computationally efficient, explainable | Sensitive to outliers, liable to overfitting, many assumptions (linearity, normality of errors). Cannot resolve nonlinear relationships in the data. |
| **Random Forest** | Scikitlearn ensemble decision tree regressor | Computationally efficient for large parameter spaces, robust to multiple non-dependent variables, allows for easy inspection of feature importance to enhance model interpretability | Computationally expensive for large datasets, lower reliability for unbalanced datasets, more complex than regression. |
| **XGBoost** | Optimized distributed gradient boosting and parallel tree boosting algorithm | Most advantages listed for Random Forest, computationally efficient for large parameter spaces, highest accuracy method in several machine learning competitions | Limited interpretability relative to other algorithms, remains liable to overfitting for small training datasets, cannot reasonably extrapolate beyond the range captured in the training data. |

the other hand, emphasizes larger errors, highlighting potential issues in model predictions. Additionally, our implementation supports a variety of other metrics which may be added if users require, including R2, residual skewness, residual kurtosis, slope, intercept, Nash-Sutcliffe Efficiency (NSE), correlation, relative variance, and bias, allowing for a thorough validation of the model across different aspects of performance and different expected error profiles.

## 2.5 Data export

As a final stage in AtsMOS, the corrected forecast variables are saved along with their metadata in the self-describing and interoperable MDF format (GEO Mountains, 2022). This final export stage has the benefit of: (i) enabling easy usage of the custom forecasts in other applications, or plotting dashboards; (ii) ensuring that the variables are saved with all necessary context for long-term archiving; and (iii) through standardized nomenclature, enabling easy comparison with other forecast datasets and external validation.

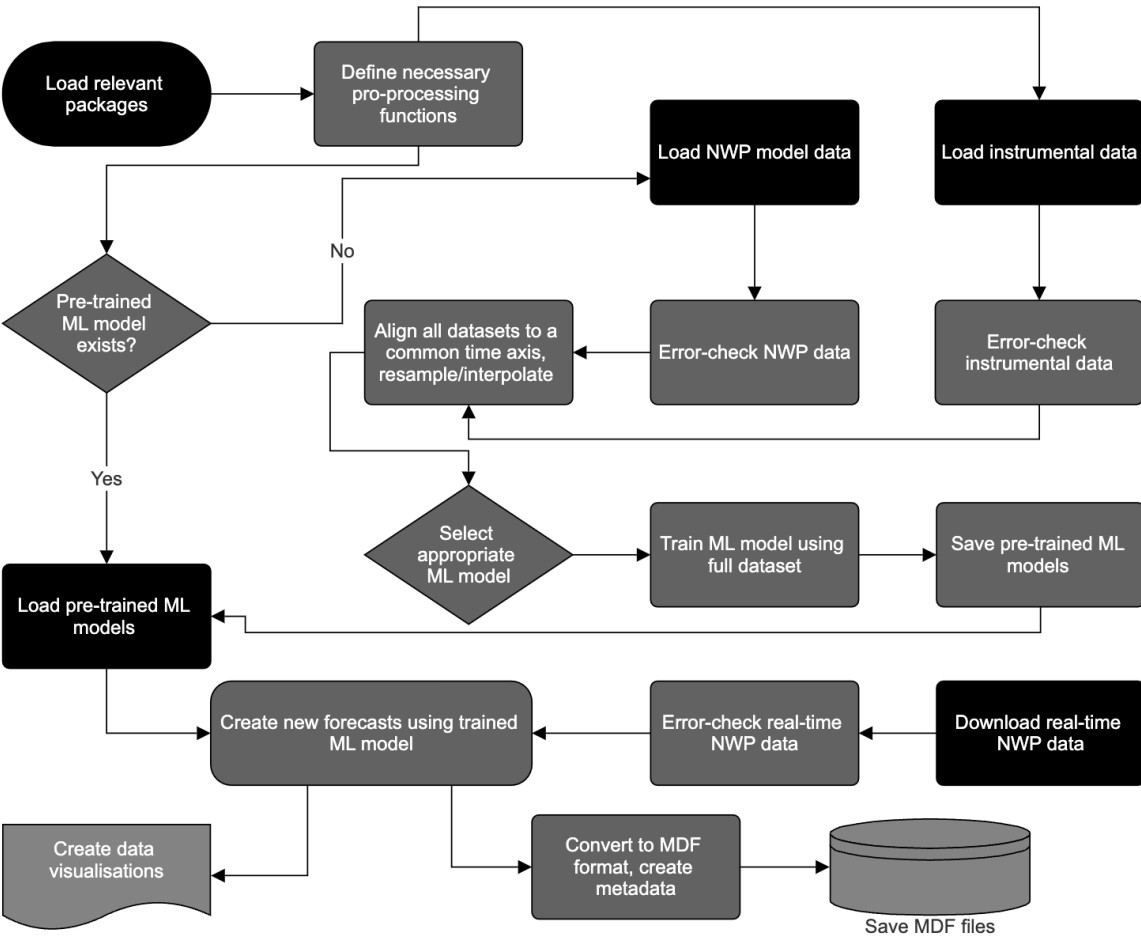

**Figure 2.** AtsMOS process workflow, described further in the documented AtsMOS jupyter notebook at zen-odo.org/doi/10.5281/zenodo.10889509.

Overall, the AtsMOS workflow is designed to be lightweight and flexible, while enhancing the predictive skill of large-scale forecasts using local observations.

## 3 Example application: Mt Everest summit meteorology

### 3.1 Background

As the highest peak on Earth, Mt. Everest sees hundreds of attempts to scale its 8850 m a.s.l summit each year. Fatalities are common, including 17 fatalities in spring 2023 (Ellis-Petersen, 2023), and an overall mortality rate of around 1 % over the past decade (Huey et al., 2020). Weather is a major contributor to these, playing a role in 25 % of deaths (Firth et al., 2008), due

to the hazard from extremely low barometric pressure (low oxygen availability) and severe cold hazard that climbers may be exposed to (Moore and Semple, 2006; Matthews et al., 2020a, 2022). The latter is sensitive to wind speed (Moore and Semple, 2011), which if high enough may also directly blow climbers off the mountain. Therefore, climbers limit their summit attempts to periods when the Subtropical Jet's retreat leaves lower wind speeds on the mountain. Therefore, accurately forecasting these periods of lighter winds is critical for minimizing the risk of climbing Mt Everest.

Whilst deciding the acceptable wind speed threshold for summit attempts is dependant on individual climbers' risk tolerance, physical considerations suggest that a human with an effective surface area ($A_p$) of 0.5 m$^2$ is at risk of being blown over if the wind force ($F$) exceeds 72 N (Hugenholtz and VanVeller, 2016; McIlveen, 2002). $F$ is related to the wind speed ($v$) according to:

$$F = \frac{1}{2}\rho v^2 A_p C_D \tag{1}$$

where $\rho$ is the air density (kg m$^{-3}$) and $C_D$ is the drag coefficient (dimensionless). Using $C_D$=0.6 from McIlveen (2002), the critical wind speed ($v_c$) yielding 72 N can be evaluated:

$$v_c = \sqrt{\frac{144}{(0.3\rho)}} \tag{2}$$

At the altitude of the highest camp (the South Col: 7,945 m a.s.l:Figure 3) on Mt. Everest's main route from Nepal – which marks the beginning of the 'death zone' – $\rho$ (which depends on temperature and pressure) is, on average, 0.52 kg m$^{-3}$, translating to $v_c = 30.3$ $m/s$ according to data from May 2019 until June 2023 (see the following section). To illustrate the use of AtsMOS for the delivery of decision-critical forecasts we therefore use a new network of Mt. Everest weather stations (see below) to develop predictions of (1) absolute wind speed; and (2) the probability of speeds exceeding both 30 $m/s$ and 20 $m/s$. The upper threshold is used to identify dangerous winds (high hazard), whilst the lower we regard as potentially dangerous (medium hazard) and hence a conservative threshold for identifying suitable weather for a summit attempt. We also showcase the flexibility of AtsMOS to directly forecast key variables such as windchill temperature and facial frostbite time.

### 3.2 Mount Everest weather data

In spring 2019, a network of five automated weather stations was installed on the Nepali side of Everest, known locally as Sagarmatha or Qomolangma, including three stations above the basecamp at Camp 2 (6464 m), the South Col (7945 m), and Balcony (8430 m; Matthews et al. (2020a)). Of these, the two highest stations: the 'South Col' (7,945 m a.s.l) and the 'Balcony' (8,430 m a.s.l) were positioned to monitor the potentially dangerous winds on the upper mountain. However, the Balcony's record is relatively short (due to wind damage), and considered unrepresentative of the upper mountain due to sheltering under common flow directions. A further station was installed at 8810 m altitude on the highest elevation exposed bedrock near the summit (the 'Bishop Rock') in Spring 2022, which is currently the highest altitude weather station in the world with publicly

available data (Matthews et al., 2022). Note that another station was installed by a Chinese team at a similar altitude on the
North side of the mountain in 2022, although its status and data availability are unknown.

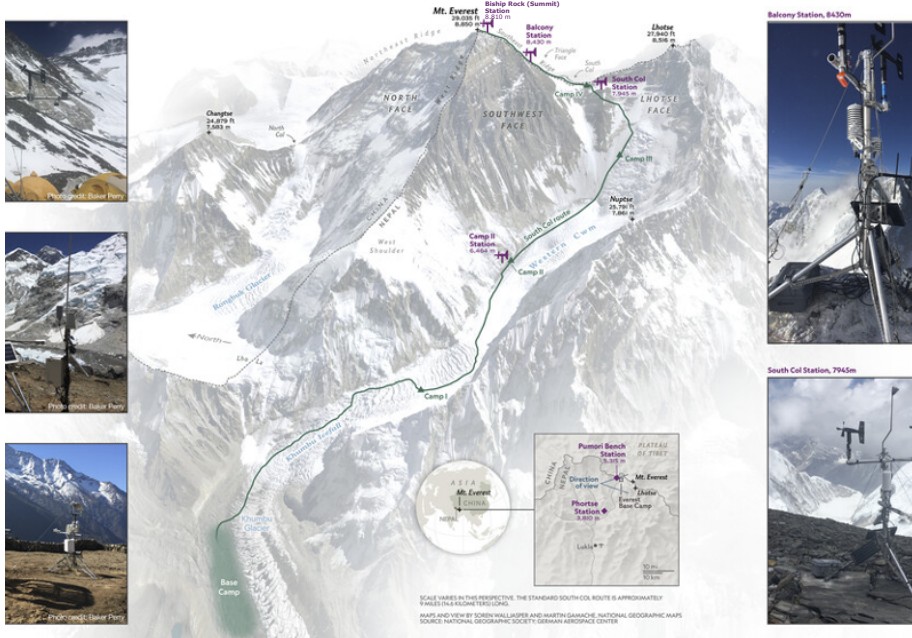

**Figure 3.** Location of the weather stations on Mt Everest. Modified after (Matthews et al., 2020b)

Three of the four weather stations (Bishop Rock or 'Summit', Balcony, and South Col) were installed with two separate wind
speed sensors. The dual-sensors were installed for redundancy in the event of one failing, but are also valuable for evaluating
the reliability of wind speed observations. Recovery of destroyed monitoring equipment showed that the wind-speed sensors
can suffer mechanical failure (breakage of the anemometer cups) and growth of rime-ice that result in incorrect measurements,
but that is not evident from a single time series. Therefore, we apply a moving-window cross-correlation between the two
sensors' time-series in the pre-processing stage of AtsMOS to identify periods of decorrelation and unreliable data. We use a
minimum correlation threshold of 0.9 measured over a 14-day window for both the mean and maximum hourly wind speed
(measured at 5-s intervals) to determine reliable data and mask out data points falling below this threshold (Figure 5 top two
rows).

Only the South Col station has a data record covering a period longer than a few months and across multiple years. While
this station is located almost 1000 m below the summit, its position at the head of the Khumbu Valley with an open westerly
aspect (the dominant wind direction) means that its wind speeds are very similar to the summit (Pearson's r-value = 0.85). We
apply the dual-sensor correlation threshold (0.9), and filter out winds with a direction outside the range $270 \pm 45$ degrees due
to the risk of topographic shielding outside this window. The lower elevation leads to a slight negative bias in wind speed at
the South Col, which is on average 18% slower than summit winds. We linearly regress the remaining South Col wind record

against the filtered summit record and use this to create a synthetic summit record. The resulting record contains just over 1 year of data, spread across two 6-month periods from 06/2019 to 01/2020 and 05/2022 to 01/2023 (Figure 5).

For the NWP component of the AtsMOS loading and pre-processing stage data were loaded and pre-processed from the Global Forecasting System (GFS) (https://rda.ucar.edu/datasets/ds084.1/) We downloaded all 10 variables: precipitation, tem-
perature, relative humidity, N-S wind speed, E-W wind speed, vertical velocity, geopotential height, absolute vorticity, cloud mixing ratio, and ozone mixing ratio. We choose to include all variables (irrespective of whether physical connections to the predictand could be identified a priori) because (i) their variations could provide insight into relevant sub-grid scale process, and (ii) the default machine learning method we select (XGBoost) is robust to overfitting and collinearity. A user-supplied list of variable names can also be supplied to AtsMOS to limit the variables used in model fitting. The data were downloaded for
a 9x9x3 data cube centred on the summit of Sagarmatha/Qomolangma, with 9 data points in each horizontal direction (from 27-29 degrees latitude and 86-88 degrees longitude at 0.25 degree spacing) and 3 vertical pressure levels (350, 400, and 450 hPa). We use the geopotential height from the three pressure levels to linearly interpolate or extrapolate all variables to a fourth vertical level, corresponding to the summit elevation at 8849 m. Finally, we calculate the horizontal and vertical gradients in the 9 variables, to further account for potential drivers of relevant sub-grid scale processes. A full list of all 172 variables
and derivatives used is in the supplement. We separately download the GFS historical archive (via the NCAR web portal) and real-time GFS forecast (programmatically from the THREDDS server - see notebook), with the former used to calibrate our data corrections and the latter used to produce corrected forecasts.

For the core processing component of AtsMOS we use (simple) linear regression and (complex) XGBoost algorithms to improve the GFS forecast for the wind speed at the summit of Sagarmatha/Qomolangma. To avoid issues with temporal
autocorrelation of training and validation data, we split our time series in half in January 2021. This test-train split provides us with 6 months of training data and 6 months of validation data from 06/2019 to 01/2020 and 05/2022 to 01/2023. We run each MOS algorithm twice, once training on data from 2019 and testing on data from 2022 and vice versa. Linear regression is applied using just the GFS model wind speed interpolated to the 8849 m summit altitude as the only predictor variable; XGBoost, on the other hand, is trained using all 172 GFS variables and spatial derivatives. We reproduce the simple and
complex MOS workflows for several GFS lead times: analysis (0 h nowcast), 24 h, 48 h, 120 h (5 day), and 240 h (10 days).

While predicting Sagarmatha/Qomolangma wind speed as a continuous variable is scientifically interesting, a categorical prediction of dangerous versus safe winds may be of more use to the majority of potential end users (Sherpas and mountaineers). We therefore employ a wind speed threshold of 30 m.s-1 to classify our synthetic wind time series into a time series of dangerous winds. We also use a second, lower threshold of 20 m.s-1 to classify potentially dangerous winds. A wind speed of
30 m.s-1 corresponds approximately to the wind speed required to blow a human off their feet at Sagarmatha/Qomolangma summit conditions (Section 3.1) We intentionally do not call winds below this threshold 'safe' as they can still be hazardous in a range of ways (including slowing ascents and increasing exposure), but they correspond to conditions during which – at least in principle – a typical climber should not be in danger of being blown from the mountain. We use the same XGBoost MOS to run the categorical forecast, using GFS lead times of 0 h (analysis), 48 h, and 240 h (10 days). For the 0 h and 48 h lead
times we forecast dangerous winds at the native GFS 6 h temporal resolution. For the 240 h (10 days) lead time, however, we

inverse the problem and classify 48 h (2 days) periods during which all winds are below the given threshold. The objective of classifying low-wind periods with a 10-day lead time is to enable earlier identification of favourable summit weather conditions and a better distribution of climbs throughout the season to prevent potentially dangerous overcrowding.

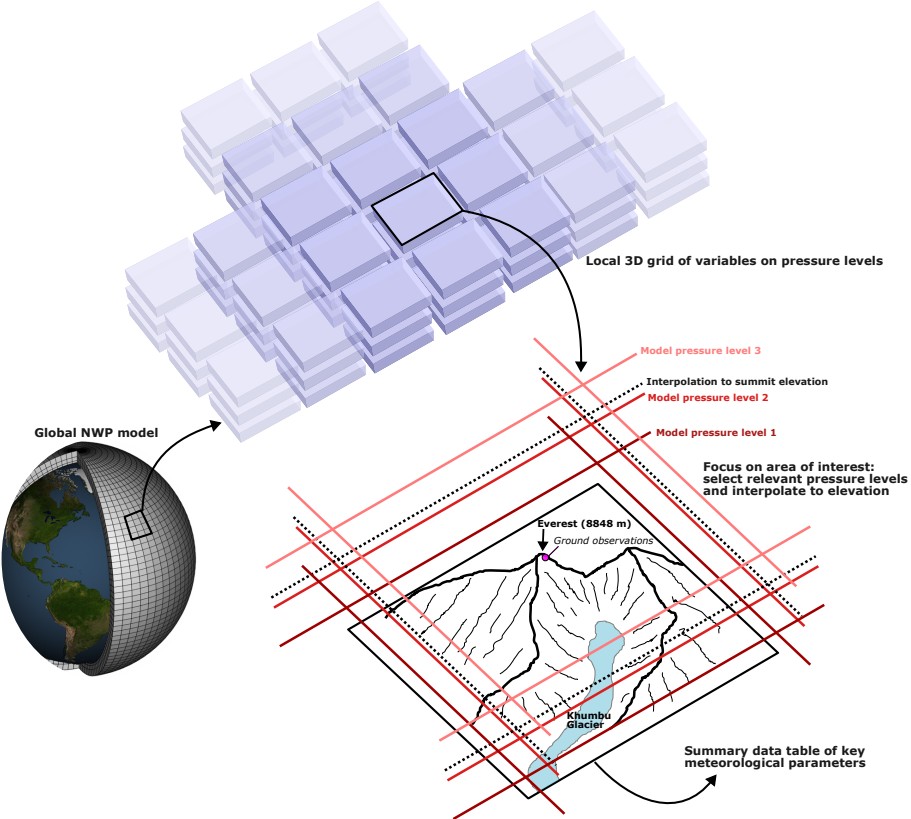

**Figure 4.** Module for extracting key meteorological parameters from the global Numerical Weather Prediction (NWP) model around a point of interest (typically, the location of the ground observations). Both horizontal and vertical derivatives are calculated from the NWP data to supplement the ML training dataset.

## 3.3 Results and model evaluation

220 We test the AtsMOS dataset by training it on data from the first period (2019-2020) and predicting data over the second period (2021-2022) and vice versa. This enables a more robust validation than random test-train splitting of the dataset, by reducing inflation of model ability caused by meteorological time series temporal autocorrelation. We evaluate three different learning techniques: simple linear regression, Random Forest, and XGBoost (Figures 5, 6, 7).

Linear regression produces a reasonable overall fit to the data (Figure 6), with a model wind speed-field measured wind
225 speed $R^2$ of 0.87, a root mean squared error (RMSE) of 10.59 m.s-1, and a mean absolute error (MAE) of 7.87 m.s-1. The Kling-Gupta efficiency of these datasets is 0.73, evaluating a combination of their correlation, relative variation, and mean

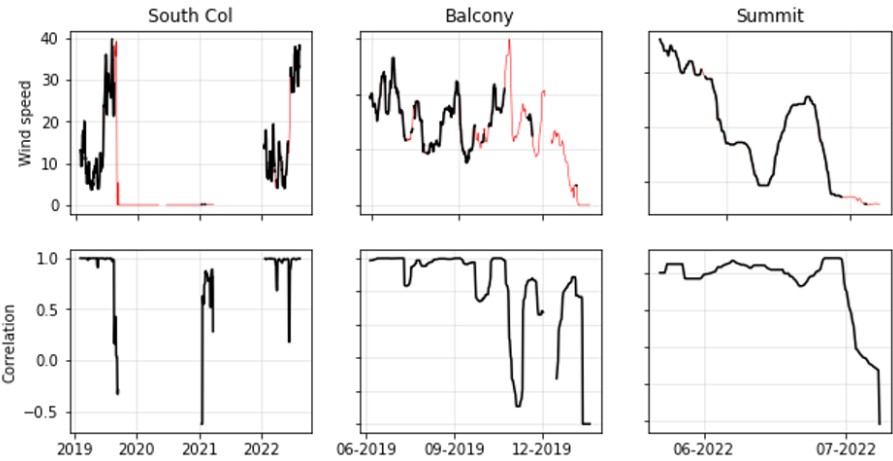

**Figure 5.** Validation of observational time series. Wind speed data shown in black (upper row) was considered reliable based on the dual-sensor correlation (lower row), while wind speed data show in red was judged unreliable and removed.

bias and with higher values reflecting a better fit (Gupta et al., 2009). In particular, linear regression successfully matches the magnitude of winds during the majority of the low-wind (monsoon) season from July to October. However, it fails to match the magnitude of the highest wind-speed events, with a clear overestimate evident.

230     Random Forest produces a good overall fit to the data, with a model wind speed-field measured wind speed R2 of 0.92 and a root mean squared error (RMSE) of 8.52 m.s-1, and a mean absolute error (MAE) of 6.33 m.s-1. The Kling-Gupta efficiency of these datasets is 0.77. There are three notable improvements of the model trained with Random Forest regression relative to standard linear regression: the estimates are more closely clustered along the 1:1 model-data line, the timings of high-wind episodes in the model better match those observed in the data, and the magnitude of high-wind peaks better matches across 235  both datasets – although a small bias towards higher winds than reality remains.

    XGBoost produces a good fit to the data, with a model wind speed-field measured wind speed R2 of 0.93 and a root mean squared error (RMSE) of 7.95 m.s-1, and a mean absolute error (MAE) of 5.97 m.s-1. The Kling-Gupta efficiency of these datasets is 0.79. The overall performance of the model trained with XGBoost is similar to that trained with Random Forest, with a slightly improved fit across all metrics. The timing of high-wind events is well predicted and, while the model still tends 240  to overestimate the magnitude of high-wind events, the bias is lower (bias score: 0.86 for XGBoost, relative to 0.84 for Random Forest and 0.81 for linear regression).

    We then apply the AtsMOS workflow on a real-time case study for the approximately two-week (384-hour) period from 20 July 2023 to 05 September 2023 using GFS forecast data as described in the methods. As well as calculating the wind speed, temperature, and precipitation, we compute forecasts of two derivative variables: wind chill temperature and facial frostbite 245  time (Moore and Semple, 2011). Both wind chill temperature and facial frostbite time are calculated based on wind speed and temperature forecasts according to the formulas of Moore and Semple (2011). Wind chill temperature reaches as low as -45

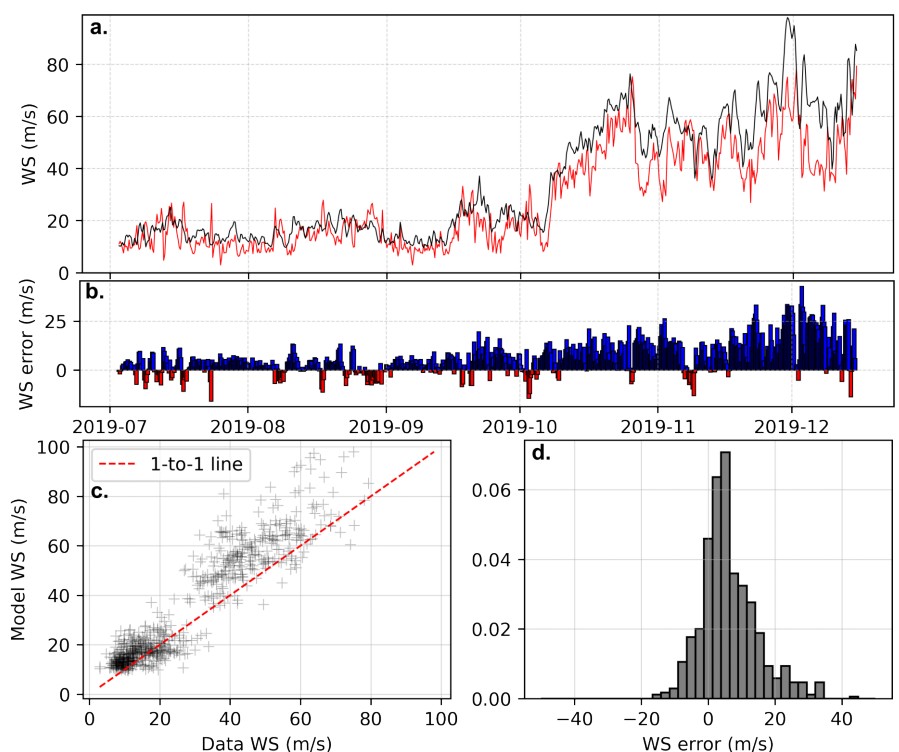

**Figure 6.** a. Observed (Red) and modelled (black) wind speed for the first observational period at the summit, with model (here, Linear Regression) training using only the second period (2019). b.,c.,d. Difference between modelled and observed wind speed shown as a difference bar chart, scatterplot, and histogram respectively. The statistics are as follows: $R^2$=0.87, Root Mean Squared Error (RMSE)=10.59, skew=0.37, kurtosis=1.25, Mean absolute error (MAE)=7.87, Nash-Sutcliffe Efficiency (NSE)=0.78 , Kling-Gupta Efficiency (KGE)=0.73, correlation=0.93, relative variance=0.81, bias=0.81.

degrees celsius on 03/09/2023, also aligning with the shortest facial frostbite time of less than 7 minutes (Figure 9). Forecast wind speeds do not exceed 20 m.s-1, but reach more than 19 m.s-1 on the night of 20-21 July 2023, with the short forecast lead reducing the uncertainty for the forecast. The facial frostbite time briefly falls below 10 minutes this night also driven by the high wind speeds, and the wind chill temperature fluctuates between -35 and -40 C – well below the air temperature (-20C), highlighting the importance of the wind speed in modulating the cold hazard, and thereby the value of computing this derived variable with AtsMOS.

## 4    Discussion and broader applicability

The AtsMOS workflow builds on advancements in machine learning and data accessibility to improve mountain weather forecasts by downscaling coarse numerical model outputs to specific locations of high value. Through a case study focusing

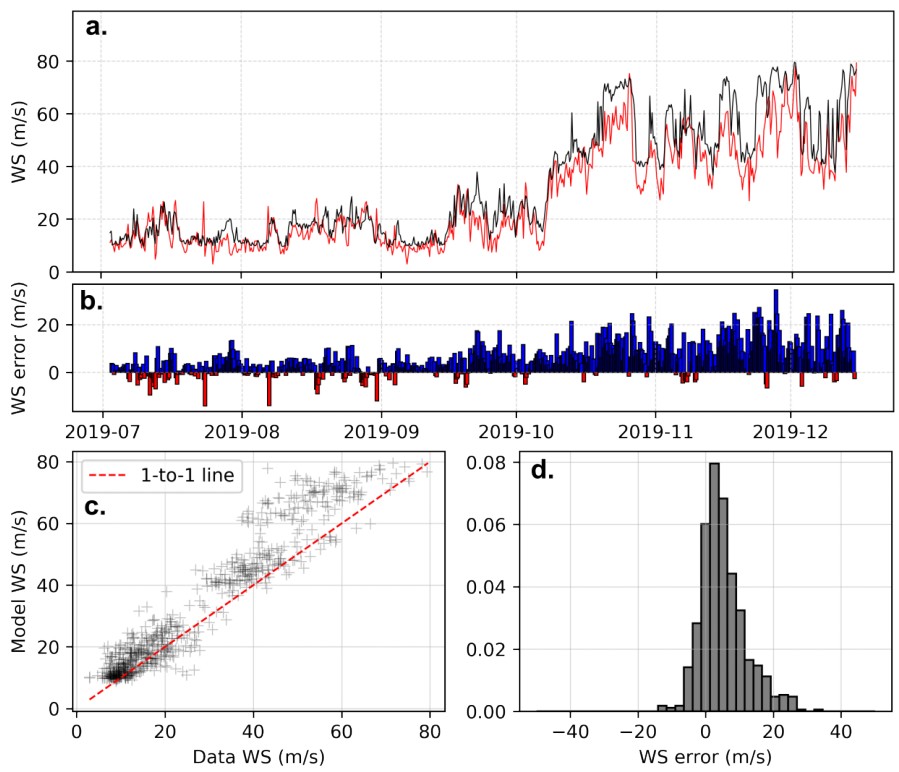

**Figure 7.** a. Observed (Red) and modelled (black) wind speed for the first observational period at the summit, with model (here, Random Forest) training using only the second period (2019). b.,c.,d. Difference between modelled and observed wind speed shown as a difference bar chart, scatterplot, and histogram respectively. The statistics are as follows: $R^2$=0.92, RMSE=8.52, skew=0.39, kurtosis=1.2, MAE=6.33, NSE=0.84, KGE=0.77, correlation=0.96, relative variance=0.84, bias=0.84.

on Mt. Everest summit meteorology, we demonstrate the effectiveness of AtsMOS in refining wind speed (and wind chill temperatures) critical for assessing risks for mountaineering. This workflow is open-source, flexible, and computationally cheap – enabling more accurate mountain weather predictions across many different environments.

Our Mt. Everest case study showcases a local application of the AtsMOS workflow, for an environment in which mountaineers and Sherpas knowingly expose themselves to potentially deadly conditions (Moore and Semple, 2006, 2011; Matthews et al., 2020b). Therefore, more precise meteorological forecasts are critical for expedition planning. First, by more accurately predicting wind speeds, our system enables expedition organizers to identify windows of potentially 'safe' (lower wind) conditions with approximately two weeks' notice, allowing the timing of trips to the upper mountain to be determined at an earlier date. This is invaluable for optimizing expedition scheduling, maximizing the likelihood of successful summit attempts, and potentially improving safety by preventing dangerous overcrowding from teams rushing to exploit weather windows at late notice. Second, the shorter lead time, and more precise AtsMOS forecasts assist in preventing climbs during times of dangerous weather, thereby enhancing safety. By providing reliable forecasts for both dangerous and potentially dangerous wind

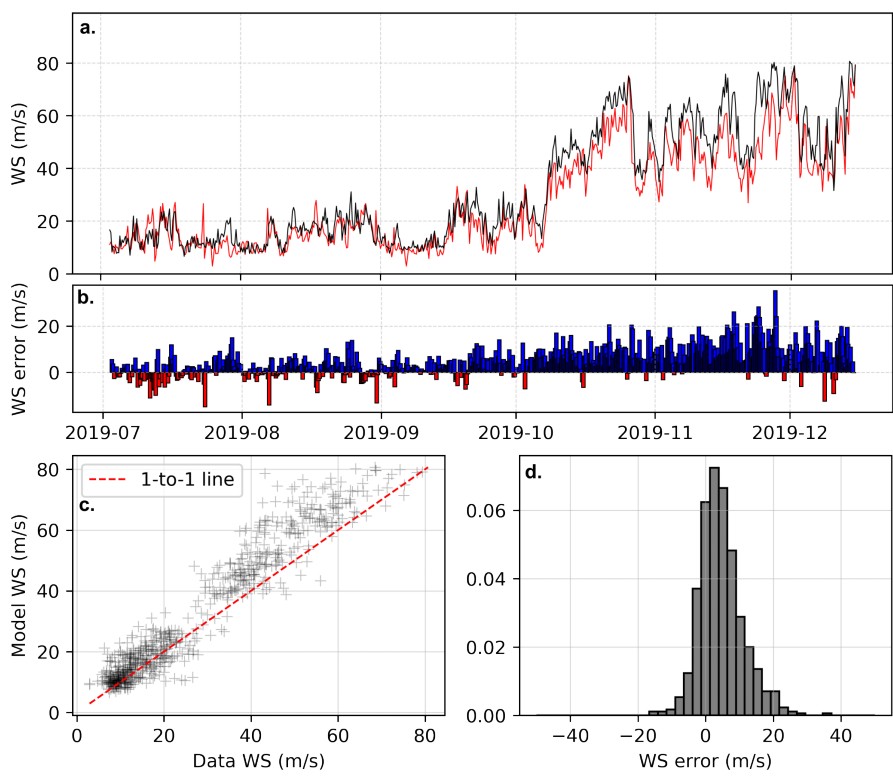

**Figure 8.** a. Observed (Red) and modelled (black) wind speed for the first observational period at the summit, with model (here, XGBoost) training using only the second period (2019). b.,c.,d. Difference between modelled and observed wind speed shown as a difference bar chart, scatterplot, and histogram respectively. The statistics are as follows: $R^2$=0.93, RMSE=7.95, skew=0.29, kurtosis=1.69, MAE=5.97, NSE=0.86, KGE=0.79, correlation=0.96, relative variance=0.85, bias=0.86.

thresholds, our workflow empowers expedition leaders to make informed decisions, avoiding ascent attempts during periods of heightened risk.

The flexible nature of our workflow enables outputs with different levels of complexity (e.g. Figure 9), ranging from binary classifications ('dangerous/safe'), raw meteorological variable forecasts (wind speed, temperature, etc.), and derivative variables (e.g. facial frostbite time). This flexibility offers a wide range of possibilities to enable expedition planners who, armed with more information, should be able to plan safer climbs, thereby reducing the risk of attempting this iconic mountain. One example is shown in Figure 9c, with 'medium' and 'high' hazard times delineating periods with high probabilities of strong

winds. The evaluation of hazard probability with AtsMOS is seen as a particularly important feature for end-users. If properly calibrated, it more clearly aligns the forecast product with decision-making. Without the MOS approach here, expedition planners would likely need to consult ensemble forecasts (e.g., the Global Ensemble Forecasting System) to produce comparable probabilities, associated with a non-trivial increase in data processing for support teams; and/or burden on the expedition planner to interpret the forecast. Of course, we also note here for AtsMOS to be used within such ensemble forecasting systems –

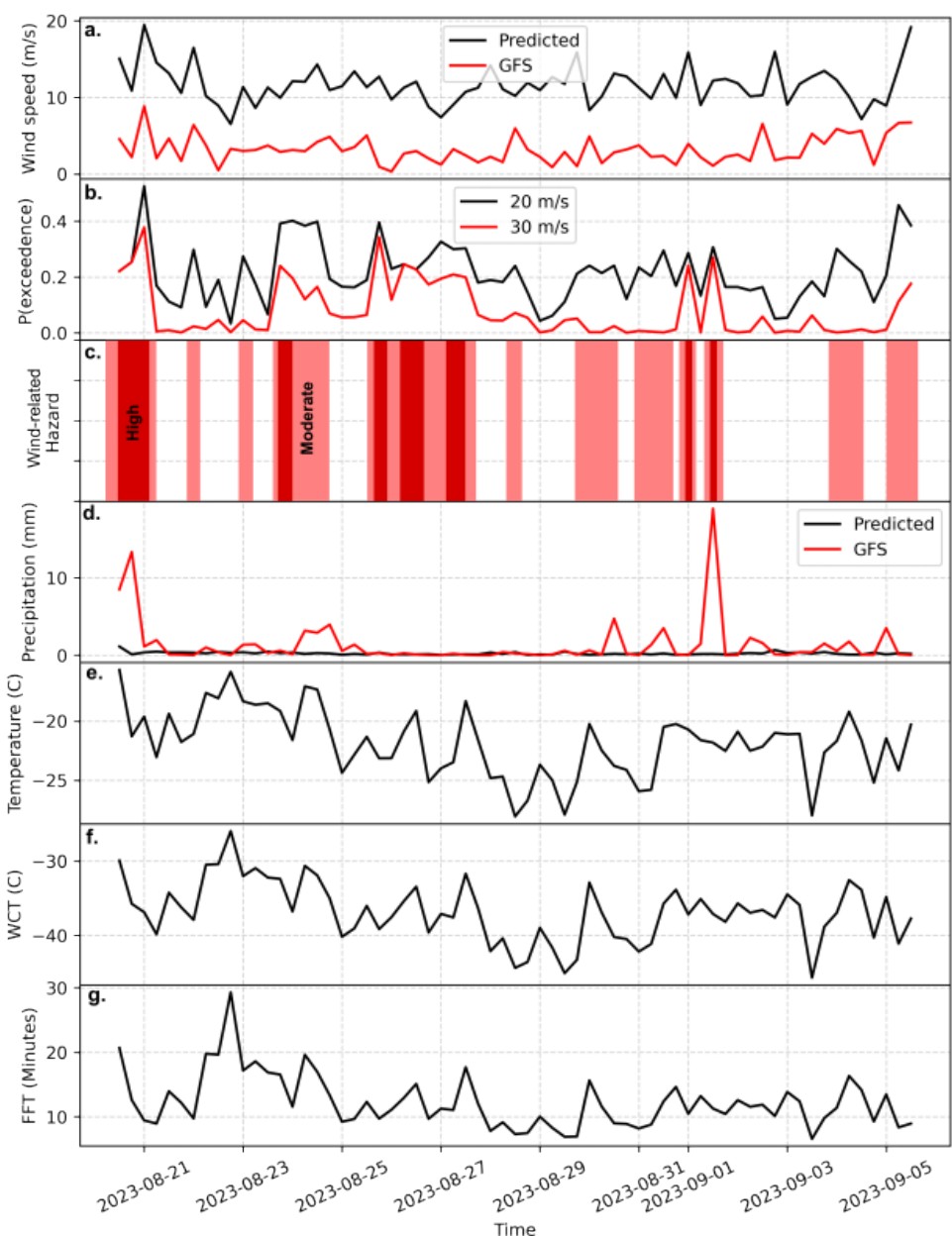

**Figure 9.** Example of real-time forecast for wind(a.), precipitation (d.), temperature (e.), as well as derivate variables of wind chill temperature (f., WCT) and facial frostbite time (g., FFT; Moore and Semple, 2011). The raw GFS estimates of wind speed and precipitation are shown in red alongside our predictions, note how the AtsMOS wind speed estimates are substantially higher and precipitation substantially higher. The probability of exceedance of different wind speed thresholds, 20 and 30 m/s, is also shown in (b.), with related hazard classification with medium hazard denoted as >20% probability of winds exceeding 20 m/s and high hazard as >20% probability of winds exceeding 30 m/s.

for example, propagating the ensemble members through the ML algorithms calibrated on the deterministic forecast to explore uncertainty more fully. This can ultimately combine the benefits of both the ensemble forecast and reduced bias from local calibration.

While the AtsMOS workflow's potential to improve local mountain meteorology forecasts is promising, it is important to acknowledge its limitations. The most significant constraint lies in the workflow's dependence on the two underlying data
sources: numerical weather prediction data and instrumental/field data. AtsMOS outputs therefore rely on the assumption that, while these datasets may contain uncertainties or noise, they both contain real and useful information about local meteorological conditions. There are a number of scenarios in which this may not be the case for either dataset, for instance, large-scale NWP models missing key local processes (leading for example, to a poor representation of convection), or sensors may become degraded and record false data (for instance, a wind sensor covered in rime-ice). This limitation is present at both the training
and prediction stages of the process. The effectiveness of the workflow is, therefore, highly dependent upon the availability and quality of ground observations, which are particularly rare in remote and high-altitude regions like Mt. Everest (Matthews et al., 2020a; Thornton et al., 2022). The applicability of AtsMOS may also be limited in regions with unique or extreme meteorological conditions not adequately captured by existing NWP models – even with the aid of machine learning to extract additional information. – which may be of concern if these regions are of particular interest for hazard mitigation. We note,
however, that the latter may be guarded against by using near real-time (i.e., lagged) observations from the telemetry-enabled weather stations (Chkeir et al., 2023).

We also highlight caution in the application of machine learning algorithms. Whilst techniques like Random Forest and XGBoost can offer enhanced predictive capabilities, they may also introduce complexities in model interpretation and require careful validation to ensure robust performance. These limitations underscore the need for ongoing refinement and validation
of the workflow to optimize its utility and effectiveness in diverse mountainous environments. One specific concern in the usage of tree-based machine learning algorithms such as Random Forest or XGBoost is that they cannot reasonably extrapolate beyond the range captured in the training data. This is a particular concern in areas with strong seasonal variation, where training on one season alone may lead to failure to produce meaningful predictions in the other season. In the case of Everest, this limitation is mitigated by having data covering the transition from low to high wind season, but in areas where this is not
possible alternative methods may need to be considered.

Another type of 'overfitting' may occur if machine learning inadvertently reproduces biases in the observations, for example, due to instrumentation errors. This challenge should be taken seriously, as the error could be systematic and dangerous. For example, if icing of wind sensors occurred preferentially in conditions of low temperature and high winds (i.e., periods of greater cold stress), the machine learning, trained on the errors, would underestimate the hazard most when it was greatest.
Such risks highlight the importance of thoroughly quality checking the observations in the pre-processing stage of the AtsMOS workflow. We note that, on Mt. Everest, the station design enables the detection of such icing through the use of redundant wind sensors (Matthews et al., 2020a, 2022). We hope that ongoing efforts to develop a Universal High Altitude Observing Platform (to enhance mountain weather monitoring worldwide) also be designed with such challenges in mind (Napoli et al., 2023). More generally, we emphasise that the AtsMOS approach to forecast improvement differs from efforts to embed ML in

NWP (e.g. Frnda et al., 2022). In this case, ML algorithms do not replace high-quality observational data; rather they emphasise the need for it and amplify any limitations of the data. By investing in data quality and instrumentation and leveraging ML alongside this, we increase our potential for accurate and actionable meteorological forecasts in mountainous regions.

In addition to ensuring the accuracy and reliability of sensor data, effective data management practices are crucial for maximizing the utility and impact of field datasets, particularly in the context of mountain meteorology. Good metadata, which
provides detailed information about the characteristics and origins of the data, is essential for understanding and interpreting observational datasets. Interoperability, where data can be integrated and exchanged across different platforms and systems with minimal barriers, becomes increasingly important when considering the generalizability of findings and methodologies to other mountainous environments. While the specific challenges and characteristics of each mountain region may vary, the fundamental principles and approaches developed for mountain meteorology research can often be applied more broadly and
insights and techniques developed in one region can inform and benefit studies in others. Promoting robust data management practices is key for both the effectiveness of individual research efforts and the broader advancement of mountain meteorology as a field.

Whilst we have demonstrated the added value of improving weather forecasts for Mt. Everest with AtsMO, we anticipate much greater benefits from this approach than just improving the safety of mountaineering expeditions. For instance, the ability
to forecast thresholds for rainfall-triggered landslides, snow avalanches, or flooding relies heavily on accurate meteorological data and predictive models. By integrating high-resolution local meteorological data from AtsMOS into early warning systems, communities can better prepare for and respond to extreme weather events, reducing the risk of casualties and damage. Furthermore, on a regional or national scale, the integration of detailed mountain meteorology datasets into larger-scale networks enhances the effectiveness of early warning systems by providing comprehensive coverage of weather patterns and potential
hazards across diverse landscapes. Improved prediction of meteorological conditions in mountainous regions has far-reaching implications for promoting the resilience and safety of mountain communities and ecosystems and is an important component of effective early warning systems for many hazards.

## 5 Conclusion

In conclusion, the AtsMOS workflow represents a computationally efficient template for downscaling numerical model out-
puts using one or a small number of field observations. The template outlines a flexible, modular workflow, for custom pre-processing of field observations or numerical weather model outputs depending on the need, and provides several possible core learning algorithms ranging from simple linear regression to more complex Random Forest and XGBoost. We explore an example application at Mt. Everest, which demonstrates its practical utility in improving the prediction of critical weather parameters for mountaineering safety. There are limitations to this approach, including reliance on high-quality sensor data
and potential biases inheritance in machine learning algorithms. Moving forward, continued research and observation network development hold promise for improving the accuracy and reliability of mountain meteorology forecasts, ultimately enhancing hazard mitigation efforts, and contributing to the resilience of communities living in these landscapes.

*Code and data availability.* The AtsMOS workflow is available at https://github.com/MaxVWDV/AtsMOS or https://zenodo.org/doi/10.5281/zenodo.108 (Van Wyk de Vries, 2024). Everest weather station data is available at https://www.nationalgeographic.org/society/everest-weather-data/.

*Author contributions.* MV and TM conceived the project and conducted the analyses with input from all authors. All authors commented on the analyses and final manuscript.

*Competing interests.* The authors declare no competing interests.

*Acknowledgements.* We acknowledge support from the Mountain Research Initiative via GEO Mountains under the Adaptation at Altitude Programme (Swiss Agency for Development and Cooperation Project Number: 7F-10208.01.02). We further thank editor Di Tian and two
anonymous reviewers for their valuable contributions to this improved manuscript. Finally, we acknowledge the crucial contributions of all sherpas, scientists, and mountaineers involved in establishing and maintaining the meteorological stations on Mt Everest.

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
