# Peer review of "At-scale Model Output Statistics in mountain environments (AtsMOS v1.0)"

_Geoscientific Model Development, 2024_

## Author Comment (AC1)

We thank both reviewers for their comments and their positive assessment of the manuscript. We have made a number of changes to our manuscript in response to the recommendations, in particular focusing on expanding and clarifying the methods section. We have added a new, more comprehensive methods flowchart, and additional methods table for the core machine learning modules, and additional details about different elements of the pre and post processing. We have also reviewed the full manuscript text and made small adjustments and corrections where recommended, which are highlighted in the 'track-changes' manuscript. We respond to all reviewer comments between the lines below.

**Reviewer 2**

This paper describes a model output statistics (MOS) approach aimed at improving weather forecasts in mountain environments. The authors focus on Mount Everest and show example usage with high elevation station data from the region. In addition to the model description, they explain their procedure for pre-processing the station data. I agree with the author's arguments on the importance of improved forecasts on Everest to improve climber safety and the need for combined NWP and in situ data to achieve this. The high altitude, telemetry equipped station network on Everest provides unique and interesting opportunities for model development targeting this goal. The example is instructive (though limited to one location) and the discussion section provides valuable context on strengths and limitations of the model. I agree with reviewer 1 regarding the need for a more detailed description of the model and I consider this the main area that should be revised/extended. I have a few additional suggestions below.

Thank you for your thorough review and constructive feedback. We appreciate your recognition of the importance of our work in improving weather forecasts for climber safety on Everest. We agree that a more detailed description of the model is necessary and will revise and extend this section accordingly. We will address the additional suggestions in detail below

General comments:

> Given the nature of the journal, I would like to see a more detailed description of the various model components. The model code is available but the documentation of the code could also be extended to help potential users get started. For the paper, I would suggest a subsection for each of the main processing modules, with particular focus on the steps in the "core processing" section of Fig. 1. This should include an explanation of the various learning techniques that were implemented.

We appreciate the feedback and have made substantial revisions to our manuscript to address these concerns. Recognizing the need for a more detailed description of the various model components, we have significantly expanded our methods section. This includes the addition of a new figure (detailing the workflow) and a table (outlining the specifics of the implemented ML models). The methods section has been reorganized into distinct subsections, each

focusing on the main processing modules, with particular emphasis on the steps in the "core processing" section of Fig. 1. We have provided detailed explanations of the learning techniques employed, such as XGBoost and Random Forest, including their parameter settings and implementation processes. Additionally, we have enhanced the documentation of the model code to assist potential users in getting started.

A subsection in the methods on the performance metrics used for model evaluation would also be beneficial.

We appreciate the suggestion and agree that a detailed subsection on the performance metrics used for model evaluation will enhance the clarity and comprehensiveness of our manuscript. We have added this subsection to the methods section ('2.4 Post-processing and validation metrics') to provide readers with a clear understanding of how we assess model performance.

Figures: Captions should be extended to fully explain the contents of the figures. Alternative visualizations for the contents of Figures 5-7 might be explored to better show the differences between learning methods.

We have modified figures 5-7 (now, figures 6-8 in the new manuscript) to better highlight the differences between the methods and capacity of the MOS technique to produce skillful forecasts. We show an example below. We have cropped the first line graph to only the region of overlap between data and model, provided an additional 'running error' plot to accompany this (b) and replaced the plot density graph with an error distribution histogram (d). Figure 8, showing the results for XGBoost, is shown below:

[Figure]

Specific comments:

Fig 4: What are the black and red lines? Add info in a legend, extend the figure caption to explain what is shown in the figure.

This caption has been extended and clarified to read "Validation of observational time series. Wind speed data shown in black (upper row) was considered reliable based on the dual-sensor correlation (lower row), while wind speed data show in red was judged unreliable and removed."

Fig 5-7: hard to see differences between the learning methods. Can this be combined to show the different model results in the same plot? What do the colors in the mesh plot (lower right panel) represent? Please explain in the caption, add a colorbar.

As discussed above, we have modified these three figures to better highlight these factors. We do not combine all three plots into 1 as this creates very small plots which are then challenging to interpret, but we have added two new plots which facilitate comparison: a running error plot and an error distribution histogram. Furthermore we provide a full range of

error evaluation metrics for this data, for instance for linear regression: "The statistics are as follows: $R^2$=0.87, Root Mean Squared Error (RMSE)=10.59, skew=0.37, kurtosis=1.25, Mean absolute error (MAE)=7.87, Nash-Sutcliffe Efficiency (NSE)=0.78 , Kling-Gupta Efficiency (KGE)=0.73, correlation=0.93, relative variance=0.81, bias=0.81."

Fig 8: Can the GFS forecast of temp and wind speed and the station data for the same time period be added for comparison? As is, the figure shows that the model outputs something and derivative values (WCT, frostbite time) can be computed. I believe more information could be added quite easily to enhance the contents of this figure.

We have added three new things to this figure:

-Comparative plots of the raw GFS forecasts for the Mt Everest summit (for wind speed and precipitation)

-The forecast probability of exceedance of our two wind speed thresholds (20 and 30 m/s).

-A classification of our forecast timeseries into 'hazard zones' based on these probability of wind speed threshold exceedances.

We hope that this clarifies this aspect of the manuscript.

[Figure]

L184 This seems to be the first mention of Random Forest. In my opinion this should be introduced in the methods section.

We thank the reviewer for highlighting this and have added a new description of the Random Forest implementation to our methods section.

L187 Kling Gupta - This has also not been mentioned previously. Consider adding a subsection in the methods addressing your performance metrics.

As described in response to your query above, we have added a new 'performance metrics' paragraph to our methods section.

Typos:

L135 "Figure X"

This should refer to (now) Figure 5. We have corrected this.

L194 "the estimated are more closely clustered" - missing word?

This was a typo and should have read "the estimates are more closely clustered". We have corrected it.

L210 "The facial frostbit time briefly falls below 10 minutes this night also driven the the high wind speeds" – typos

We thank the reviewer for highlighting this issue and have corrected it to "The facial frostbite time briefly falls below 10 minutes this night also driven by the high wind speeds […]"

---

## Author Comment (AC2)

We thank both reviewers for their comments and their positive assessment of the manuscript. We have made a number of changes to our manuscript in response to the recommendations, in particular focusing on expanding and clarifying the methods section. We have added a new, more comprehensive methods flowchart, and additional methods table for the core machine learning modules, and additional details about different elements of the pre and post processing. We have also reviewed the full manuscript text and made small adjustments and corrections where recommended, which are highlighted in the 'track-changes' manuscript. We respond to all reviewer comments between the lines below.

**Reviewer 1**

The paper introduces an MOS (Model Output Statistics) method using artificial intelligence (machine learning) suitable for at-scale rapid deployment, for correcting deviations of numerical weather prediction (NWP) in complex mountainous areas. It uses the Mount Everest climbing meteorological service in the Himalayas as a pilot study to validate the method's feasibility. The paper also discusses the advantages, potential issues, and risks of this method.

We thank the reviewer for these comments and have responded between the lines below.

 The main review comments and suggestions are as follows:

1. As a manuscript intended for publication in GMD (Geoscientific Model Development), there needs to be a very detailed description of the described technical methods and model. In this manuscript, the technical details of AtsMOS need further refinement and organization. It is preferable to provide a more detailed flowchart than Figure 1, or to add detailed sub-flowcharts for each module, accompanied by text descriptions, especially for the implementation process, parameter settings of XGBoost and RF, etc., to enhance the practical reference value of this open-access paper.

We agree with this comment and have modified our manuscript so as to clarify these aspects. We acknowledge the need for a more detailed description of the technical methods and model. We have substantially expanded our methods section, adding a new figure (detailed workflow) and table (details of ML models implemented). We have divided this section into different parts and added further details where needed. Finally, we have added further references to our documented jupyer notebook, which we view as a parralel resource to the manuscript for users aiming for an in-depth understanding our our methods. These additions will ensure that readers have a comprehensive understanding of the AtsMOS workflow and can effectively replicate and apply the methodology in their own research.

2. The paper only conducts simulated comparative analysis and verification based on observations and forecasts of Mount Everest in the Himalayas. On the one hand, for the verification of weather transition stages (rapid temperature decrease or increase, rapid increase or decrease in wind speed), a detailed analysis is needed. On the other hand, if possible, more experiments and comparative analyses can be conducted with richer observations and AtsMOS forecasts in other mountainous regions around the world (such as the European Alps, the Rocky Mountains in the United States) to strengthen the reliability and universality validation of this method.

Thank you for your insightful comments. We chose to focus on Mount Everest as a pilot study due to its extreme meteorological conditions and clear need for skilled forecasts, which provide a useful testbed for validating the AtsMOS workflow. Future work conducting additional case studies in other mountainous regions, such as the European Alps or the Rocky Mountains, would indeed provide further validation of our method. For this manuscript we prioritise a thorough analysis of a single, challenging environment and reserve the application of AtsMOS to other mountainous regions for future work. While further validation is always valuable, we believe that the Mt Everest case study presented provides sufficient information to demonstrate that the technique is viable and is of sufficient interest to include in this GMD paper.

3. The textual presentation of the paper needs to be more rigorous. For example, some abbreviations need to be provided in full, or a list of abbreviations can be provided at the end of the paper. Sentence expressions need to be more rigorous, and writing needs to be standardized (such as subscript and superscript issues, unit measurement issues, meteorological professional expression issues, etc.).

Thank you for your suggestions regarding the textual presentation. We have thoroughly reviewed the manuscript to ensure that all abbreviations are provided in full upon first use. Additionally, we have standardized the writing to address issues with subscript and superscript formatting, unit measurements, and meteorological professional expressions where these could be identified. We hope that these changes improve the overall clarity of the manuscript.

4. For Figures 5-7, it is recommended to extract the data segment that simultaneously includes observation and forecast results and redraw clearer graphs (or add a curve showing the difference between the two ones). The current figures do not clearly show the specific differences.

We thank the reviewer for these suggestions, which we have implemented in the new version of these figures. We both cropped the first line graph to only the region of overlap

between data and model, and provided an additional 'running error' plot to accompany this (b). As noted below, we also replaced the plot density graph with an error distribution histogram (d). We hope that these modified figures 5-7 (now, figures 6-8 in the new manuscript) better highlight the differences between the methods and capacity of the MOS technique to produce skillful forecasts.

Figure 6, showing the results for the linear regression:

[Figure]

5. The titles of all figures need to be further refined to increase clarity.

Thank you for your suggestion to refine the titles of all figures. We have revised the figure captions to increase clarity and ensure they succinctly describe the content of each figure.

6. The description of Kling-Gupta efficiency needs to be clarified. This evaluation index is mainly used in hydrology. Whether it is suitable for this work should be clearly explained.

We acknowledge the need to clarify the use of Kling-Gupta Efficiency (KGE) which, while traditionally used in hydrology, is a robust metric that combines correlation, bias, and variability, making it well-suited for evaluating the performance of meteorological models as well. We have added a new paragraph in our methods section on evaluation metrics to clarify our use of KGE and to explain its relevance and applicability to our study. Additionally, we continue to use other common metrics such as MAE and RMSE.

7. An interesting question is whether further verification and comparison can be conducted for forecast results similar to Figure 8. If the Everest climbing team(s) or guides have records or carry instruments with similar data, such verification and comparison can be conducted. I believe under such extreme geographical conditions in Everest, this serves as a meaningful validation and assessment of the AtsMOS method.

Thank you for your suggestion. Unfortunately, the Everest climbing teams or guides do not currently carry instruments that provide data comparable to our forecasts. However, we agree that such verification would be highly valuable. In the discussion section, we will include a recommendation that climbers in extreme locations like Everest report simple yes/no feedback on climbable conditions, if not more detailed meteorological information. This feedback could enhance forecast validation and assessment. The flexibility of our ML system allows it to predict such binary 'climbable/not climbable' outcomes from the GFS or other forecasting systems, bridging the gap between complex data and practical decision-making for climbers.

---

## Author Response (AR2)

**Response to minor reviews: At-scale Model Output Statistics in mountain environments**

Editor Comments

Reviewer 2 has made some recommendations to improve the clarity of the figures and texts. Based on the assessment, there are also many places in the jupyter notebook need to be clarified or modified to ensure clarity and functionality of the model, which is particularly important to the readers of Geoscientific Model Development.

Thank you for the comments and suggestions. We appreciate the feedback from Reviewer 2 regarding the clarity of figures and the Jupyter notebook. We will address these points to ensure that both the manuscript and the accompanying code are as clear and functional as possible. We respond to the specific comments provided by Reviewer 2 between the lines below.

Reviewer 1

The manuscript has been thoroughly revised and improved based on the comments and suggestions from the two reviewers. No further revisions are required.

Thank you for your positive feedback and for acknowledging the improvements made to the manuscript. We appreciate your thorough review and are glad no further revisions are required. Your comments have greatly contributed to the refinement of our work.

Reviewer 2

I thank the authors for their responses and the revisions to the manuscript. The methods section has been expanded and more comprehensively describes the workflow. I have some minor comments on the figures, which I feel could still be modified a bit to improve clarity.

I also added some suggestions related to the jupyter notebook since the authors repeatedly point to this for additional documentation of the workflow. I am not sure if or to what extent code is supposed to be assessed during the review. I assume it is up to the authors whether or not to implement these points and it won't be a factor for the decision making in the review process.

Thank you for your thoughtful review and for recognizing the improvements made in the revised manuscript. We appreciate your continued feedback and have addressed your minor comments as follows:

Manuscript:
Fig 2: "Define necessary pro-processing functions" → typo, pre-processing

We have corrected the typo from "pro-processing" to "pre-processing."

Fig 6, 7, 8: The captions are improved and the figures are overall easier to follow. However, I would still recommend adding legends to the panels so that the figures can be understood in a general sense without the captions → add legend for red and black line to panel a) stating what the lines are, same for the other panels.
Since the aim of these figures seems to be comparing the different models, I am also wondering if the comparison between the three modeling approaches would be easier if the figures were divided by visualization type rather than model? The model output could be directly compared in multi-panel plots showing one kind of visualization for all three models.

We have added legends to panel a to ensure the figures can be understood independently of the captions. Additionally, we considered your suggestion to compare model outputs by visualization type rather than model. However, after careful consideration, we believe the current layout provides a clearer comparison for the readers, with the different 'difference' indicators more meaningful when adjacent to each other.

Fig 9: caption is missing explanations of panel b and c. Why not add the GFS data to the temperature, WCT and FFT panels? WCT and FFT can be calculated from GFS wind and temp and since you stress the importance of these variables you might show how they vary between GFS and your MOS approach. The irregular ticks on the x-axis seem unusual and don't add information (2 day tick spacing except for Aug 31 and Sep 1). Perhaps this could be homogenized if you have ticks for each day and tick labels every second day or some such solution.

We have revised the caption to include explanations for panels b and c, and homogenized the x-axis ticks to a daily frequency for consistency. We have also added GFS data to the temperature, WCT, and FFT panels to demonstrate the variation between GFS and our MOS approach.

L69: "synchronise measurement and NWP measurement measurement timings"
Typo, and NWP output is not a measurement. I suggest rephrasing.

We have rephrased the sentence to "synchronise measurement and NWP output timings."

L223 I think "Figures 5, 6, 7" should be 6, 7, 8 ?

Thanks, we have corrected the reference to the figures from "Figures 5, 6, 7" to "Figures 6, 7, 8."

L271 ('dangerous/safe') you say in line 211 that you do not call wind below the threshold "safe", yet here you do. I suggest checking the manuscript for consistency regarding this.

We have reviewed the manuscript for consistency regarding the terminology "dangerous/safe", and modified this to read as 'dangerous/potentially safe' instead.

Jupyter notebook:
I downloaded the files on https://zenodo.org/records/10889510 and opened the Jupyter notebook. The Readme file states "Details are in the commented jupyer notebook 'full_workflow.ipynb'". This does not exist in the folder, I believe the relevant file is called AtsMOS_workflow.ipynb
I struggled to identify which mdf package was used. A requirements.txt file or similar to make it easier to install all dependencies would be helpful.
The documentation in the notebook, which the authors refer to in the manuscript, consists mainly of comments in the code. The comments have a variable level of explanatory detail. Some functions have minimal and not very instructive comments, (e.g.: "def get_random_random_function_function(): # says what it does, does what it says"), some have no comments (eg def contiML_XGB(full,f0,real)), some have detailed comments that explain what happens in the function. This might be reworked for consistency but, again, I am unsure what the journal standards are regarding code documentation.
This is a personal preference, doesn't matter for functionality and is likely not relevant for the review process, but I find the mix of plotting and processing functions in the very large main block of code hard to follow. I would suggest breaking this up into separate cells for different function types (e.g., plotting, core-processing, …) and adding some more markdown explanations.
I think it would be helpful if the code could be linked more clearly to the workflow shown in Fig 1 and Fig 2 in the manuscript. I can see what is where but re-structuring the cells a bit and adding more markdown to clearly point out e.g. where selections are made (diamond shaped boxes in Fig 2) would probably help users find their way through it. The three modules mentioned in the manuscript (Section 2.1, loading and pre-processing, core-processing, post-processing) are not very clearly apparent as separate parts of the code. Rather, most functions are gathered in one long cell and they are called later on. As above - this does not affect functionality and may not be relevant for the review process, but I feel it might be re-organized somewhat to better match the text descriptions.

We appreciate your detailed feedback on the Jupyter notebook, and we will make these changes to improve its usability and alignment with the manuscript. The correct notebook is indeed "AtsMOS_workflow.ipynb," and we have updated the Readme file accordingly. In particular, we have removed some of the functions with fewer comments which are not necessary (were previously used for some tests), and added some additional comments throughout the jupyter notebook. We choose to keep the functions at the head of the code as several are used throughout in different sections, but we hope that the additional comments, combined with the ability to search through this code, will enable them to be linked up successfully. The most up-to-date code is available in the github and Zenodo repositories. Our objective is for this code to be easily re-usable by other researchers in their own field sites.

Thank you again for your constructive comments. We hope these revisions will significantly enhance the clarity and functionality of both the manuscript and the accompanying code, and that it is now ready for publication.